# Population genetic models of GERP scores suggest pervasive turnover of constrained sites across mammalian evolution

Christian D. Huber [ID]¹, Bernard Y. Kim [ID]², Kirk E. Lohmueller [ID]³,⁴,⁵*

1 School of Biological Sciences, University of Adelaide, Adelaide, South Australia, Australia, 2 Department of Biology, Stanford University, Stanford, California, United States of America, 3 Department of Ecology and Evolutionary Biology, University of California, Los Angeles, California, United States of America, 4 Interdepartmental Program in Bioinformatics, University of California, Los Angeles, California, United States of America, 5 Department of Human Genetics, David Geffen School of Medicine, University of California, Los Angeles, California, United States of America

* klohmueller@ucla.edu

**Data Availability Statement:** All data analyzed in the present manuscript are already publicly available. GERP scores can be found at

## Abstract

Comparative genomic approaches have been used to identify sites where mutations are under purifying selection and of functional consequence by searching for sequences that are conserved across distantly related species. However, the performance of these approaches has not been rigorously evaluated under population genetic models. Further, short-lived functional elements may not leave a footprint of sequence conservation across many species. We use simulations to study how one measure of conservation, the Genomic Evolutionary Rate Profiling (GERP) score, relates to the strength of selection ($N_e s$). We show that the GERP score is related to the strength of purifying selection. However, changes in selection coefficients or functional elements over time (i.e. functional turnover) can strongly affect the GERP distribution, leading to unexpected relationships between GERP and $N_e s$. Further, we show that for functional elements that have a high turnover rate, adding more species to the analysis does not necessarily increase statistical power. Finally, we use the distribution of GERP scores across the human genome to compare models with and without turnover of sites where mutations are under purifying selection. We show that mutations in 4.51% of the noncoding human genome are under purifying selection and that most of this sequence has likely experienced changes in selection coefficients throughout mammalian evolution. Our work reveals limitations to using comparative genomic approaches to identify deleterious mutations. Commonly used GERP score thresholds miss over half of the noncoding sites in the human genome where mutations are under purifying selection.

## Author summary

One of the most significant and challenging tasks in modern genomics is to assess the functional consequences of a particular nucleotide change in a genome. A common approach to address this challenge prioritizes sequences that share similar nucleotides

hgdownload.cse.ucsc.edu/gbdb/hg19/bbi/All_
hg19_RS.bw.

**Funding:** This work was funded by the National
Institutes of Health (https://www.nih.gov/) grant
R35GM119856 to KEL. The funders had no role in
study design, data collection and analysis, decision
to publish, or preparation of the manuscript.

**Competing interests:** The authors have declared
that no competing interests exist.

across distantly related species, with the rationale that mutations at such positions were
deleterious and removed from the population by purifying natural selection. Our manu-
script shows that one popular measure of sequence conservation, the GERP score, per-
forms well at identifying selected mutations if mutations at a site were under selection
across all of mammalian evolution. Changes in selection at a given site dramatically
reduces the power of GERP to detect selected mutations in humans. We also combine
population genetic models with the distribution of GERP scores at noncoding sites across
the human genome to show that the degree of selection at individual sites has changed
throughout mammalian evolution. Importantly, we demonstrate that at least 80 Mb of
noncoding sequence under purifying selection in humans will not have extreme GERP
scores and will likely be missed by modern comparative genomic approaches. Our work
argues that new approaches, potentially based on genetic variation within species, will be
required to identify deleterious mutations.

## Introduction

One of the largest challenges in modern medical and population genetics is determining the
phenotypic and fitness consequences of a particular mutation. Genome-wide association stud-
ies (GWAS) have implicated hundreds of loci across the genome for controlling many traits
[1]. However, finding causal variants at these loci has remained challenging due to the statisti-
cal correlations between markers (linkage disequilibrium) and by the fact that most GWAS
hits fall in noncoding regions of the genome with little obvious function [2]. Knowledge of the
particular causal variant(s) is an important goal, as it will improve risk prediction and enable a
more detailed understanding of the biological mechanism behind how the variant influences
the trait. In population genetics, there is tremendous interest in understanding how much of
the genome is under selection and the types of mutations underlying much of the phenotypic
variation and adaptation in different species. Further, studies have aimed to precisely quantify
the amount of deleterious variation segregating in populations to assess the role of population
history at influencing deleterious variation and for determining whether small population size
could lead to an accumulation of deleterious variants, potentially causing a mutational melt-
down and extinction [3–6].

One popular way to assess which mutations in a genome may be biologically functional and
affect fitness is to examine the extent to which nucleotides are conserved across evolutionarily
distant taxa. Sites showing a deficit of substitutions across many lineages are thought to be
functionally important and subject to purifying selection. Sites showing a larger number of
substitutions are thought to be evolving at a neutral rate and would be less likely to be func-
tional or under purifying selection. A number of statistical approaches have been developed to
find these sites in the genome showing conservation across disparate species [7–14]. Addition-
ally, this concept has been used in several annotation tools such as SIFT, PolyPhen, and
CADD scores to predict which mutations are likely to be deleterious [15–18].

One particular comparative genomic approach that has received widespread use is the
Genomic Evolutionary Rate Profiling (GERP) score [19,20]. The GERP score is defined as the
reduction in the number of substitutions in the multi-species sequence alignment compared to
the neutral expectation. For example, a GERP score of 4 would mean there are 4 fewer substi-
tutions at a particular site than what is expected based on the neutral rate of evolution across
the phylogeny. As such, the GERP score is a measure of sequence conservation across multiple
species. However, GERP scores have been commonly used in evolutionary genomic studies as

a measure of the strength of selection acting on derived mutations segregating within a species. In these applications, it is assumed that mutations that appear at sites that are highly conserved across many species are deleterious and thus contribute to genetic load within a species. Quantitatively, for each segregating site within a species, the GERP score is assigned to the derived mutation segregating at that site. For example, Schubert et al. [3] studied patterns of deleterious mutations in wild and domesticated horses. They computed a GERP score load for each horse which was the average GERP score over all derived variants within that individual. They found an increase in the GERP score load in the domesticated horse, arguing that domestication has led to an increase in deleterious variation. Henn et al. [5] used GERP to assess the fitness impact of amino acid changing mutations in humans. They defined mutations with GERP scores 4–6 to have "large" deleterious effects, corresponding to a selection coefficient of $10^{-3}$, and observed an increase in the number of these derived deleterious alleles in non-African populations. They also reported that the GERP scores summed over all sites within an individual were higher in an average Mayan Native American genome compared to an average San Sub-Saharan African genome. Marsden et al. [4] used GERP to identify deleterious amino acid changing mutations in dogs and wolves and found an increase in deleterious mutations (GERP>4) in dogs and that dogs have a higher summed GERP score across all amino acid changing variants as compared to wolves. Lastly, Valk et al. [6] found that, across a range of mammals, species with historically low population size and low genetic diversity have a lower average GERP score of the derived allele than species with large population sizes, suggesting that purging of deleterious alleles reduces genetic load in small populations in the long term.

While GERP scores have been extensively used in medical and population genetics, some challenges remain. First, the studies described above partition the GERP scores in a coarse fashion to reflect the underlying deleterious selection coefficient. Those mutations with a higher GERP score were assumed to have a more deleterious selection coefficient. However, the accuracy of attributing GERP scores to particular fitness effects remains unclear. GERP scores may not provide quantitative evidence of the strength of selection because any deleterious mutations that have a scaled selection coefficient of $N_e s < -2$ will not accumulate as substitutions [21–24]. Below this value, neither weakly nor strongly deleterious mutations will accumulate as substitutions, and it may thus not be possible to distinguish between them using comparative genomic data [23]. Second, most conservation-detection methods assume constant selection pressures across all branches of a phylogeny [8]. Any sort of lineage-specific selection, or turnover of functional sequence (i.e. a sequence has a specific regulatory role in one lineage, but does not in another lineage), could potentially be missed by these comparative genomic approaches. Recent evidence has suggested a fair amount of turnover of functional sequence in the noncoding regions of the human genome [25, but see 26]. Lastly, it was shown that the power of comparative genomics methods to detect sequences under selection can be maximized by selecting optimal subsets from a larger set of species [27]. However, the optimal subset of species to maximize the performance under different selection and turnover scenarios remains unclear. This is especially prescient in light of recent projects aimed at increasing the numbers of sequenced genomes across species [28,29].

Finally, the extent of the human genome under purifying selection has remained under vigorous debate. Early comparative genomic studies suggested that at most 15% of the genome was under selection [9,20,30–33]. However, biochemical studies conducted by ENCODE have suggested that up to 80% of the genome shows activity in at least one biochemical assay [34]. It may be possible to reconcile these estimates by noting that they measure different processes—functional assays assess whether the nucleotide has biochemical activity, but this activity may not necessarily be related to fitness [35,36]. As such, mutations at biochemically active sites may not have an evolutionary impact and thus could appear to be neutral in comparative

genomic approaches. Further, there has been evidence from comparative genomic studies of turnover of sequence subjected to purifying selection [25,30,33,37,38]. This could occur in a number of ways. First, sequences may have a biological function in some species and not others due to changes in regulatory architecture across species [39]. Second, even if the regulatory region retains biological function over long evolutionary times, selection coefficients of mutations at particular sites could change over time due to epistatic effects with other mutations [40]. Rands et al. suggest that the evolutionary history of the human genome has been highly dynamic, with only 25% of the elements under purifying selection in humans having maintained constraint in mouse [25,30]. Other studies have suggested that more recent evolutionary turnover has had little impact on the functional content of the genome [26]. Thus, it remains an open question as to how much of the genome is under purifying selection and the amount of turnover of functional sequence that is occurring.

Here we conduct realistic simulations under population genetic models of purifying selection to assess the performance of GERP scores under different scenarios. We first evaluate whether GERP scores can provide reliable estimates of selection coefficients at individual coding mutations. We then assess the extent to which sequence turnover affects the ability of GERP to identify selected sequences at noncoding sites. Lastly, we estimate that at least 4.51% of the noncoding portion of the human genome is under purifying selection and that mutations at most of these noncoding sites have not been under selection throughout all of mammalian evolution. Our results point to several important limitations to using comparative genomic approaches for determining the fitness effects of individual mutations and add to the growing literature arguing for using polymorphism data for assessing present-day amounts of selection within species [23].

## Results

### Estimating the strength of selection for mutations at coding sites from GERP scores

We begin by examining how the GERP score behaves under different amounts of purifying selection. The GERP score is a measure of the decrease in the number of substitutions or fixations relative to neutrality. Thus, it is directly related to the probability of fixation of deleterious mutations as derived by Kimura [21]. The ratio $\omega$ of the substitution rate under selection relative to the neutral substitution rate depends on the compound parameter of the product of haploid effective population size $N_e$ with the selection coefficient $s$ [22,24]:

$$\omega = \frac{2N_e s}{1 - e^{-N_e s}} \tag{1}$$

It is assumed that fitness of an individual with a deleterious mutation is reduced by a factor of 1-$s$ in the heterozygous state. We then simulate sequences under this model of substitution along a phylogenetic tree of 36 mammalian species (see Methods). By using the GERP++ software for estimating the number of substitutions that have accumulated at each site, we can compute a GERP score. Examining the relationship between the GERP score as a function of the strength of selection, we see that more deleterious mutations tend to have more positive GERP scores (Fig 1A). For mutations that are moderately deleterious, $N_e s < -5$, GERP scores are >4, suggesting that GERP can distinguish these mutations from neutral ones. Nearly neutral mutations, however, (-2<$N_e s$<0) have a broad distribution of GERP scores, with some being >4 and others being <0. GERP scores < = 0 suggest effectively neutral evolution (-1<$N_e s$<0).

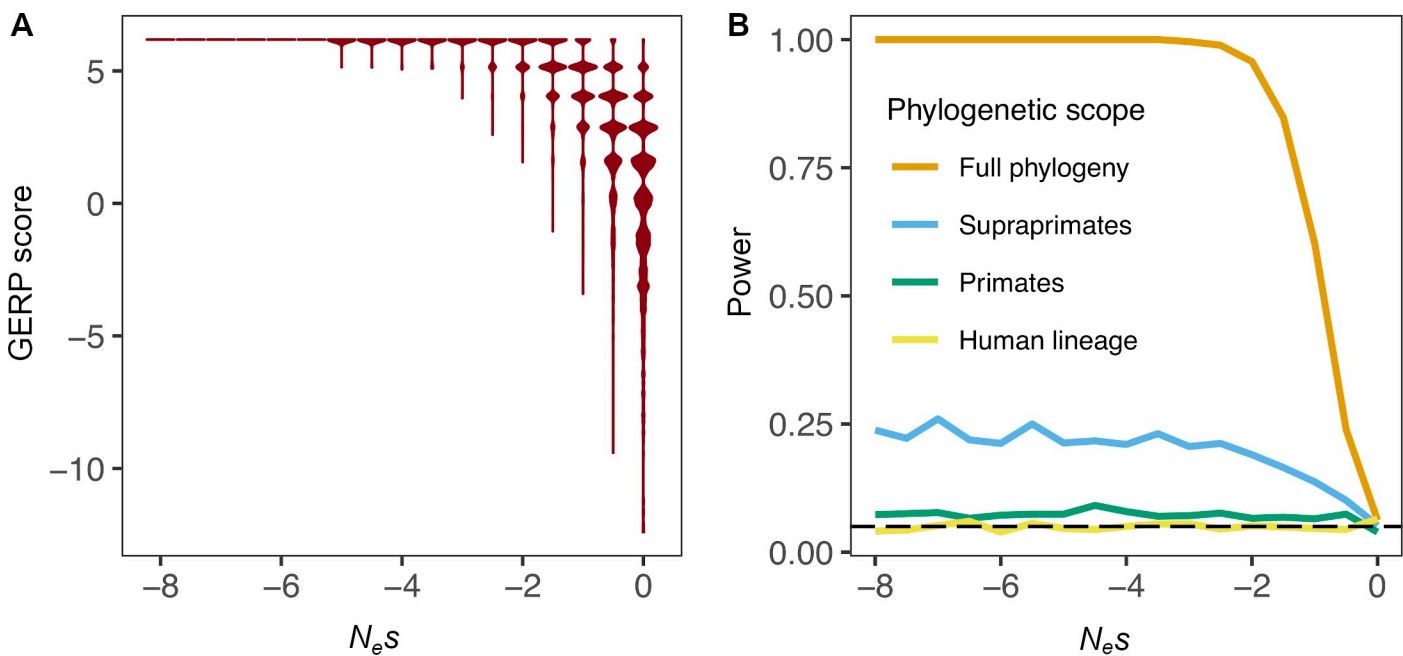

**Fig 1. GERP scores as a function of the strength of purifying selection ($N_e s$).** (A) Violin plots of simulated GERP scores on a 36 species phylogeny assuming $N_e s$ values from 0 to -8 in steps of 0.5. (B) Power of GERP to detect purifying selection, at a given selection strength shown on the x-axis. GERP scores were computed using the entire phylogeny of 36 mammalian species, but purifying selection only occurred in the phylogenetic scope shown in the legend.

Importantly, GERP scores are often used to assign selection coefficients to individual variants [5,41–43]. Fig 1A shows that the largest GERP score (GERP = 6.18, i.e. zero predicted substitutions) can be generated by weakly deleterious mutations as well as very strongly deleterious mutations. For example, both sites with weakly deleterious mutations (e.g., $N_e s$ = -4) and sites with strongly deleterious mutations (e.g., $N_e s$ = -1000) lead to the largest GERP score with high probability (98.2% and 100%, respectively). Thus, observing the largest possible GERP score for a given alignment is not very predictive of the strength of purifying selection. The reason for this is that both weakly and strongly deleterious mutations have a low probability of fixation. As such, mutations with either strength of selection will have a similar GERP score—namely, a large one because all or most substitutions were removed by purifying selection. Smaller GERP scores, however, are only compatible with weak selection or neutrality ($N_e s > $ -4).

While GERP may not be very useful at identifying the selection coefficient at individual mutations, it may be able to distinguish sites where mutations are neutrally evolving from those where deleterious mutations occur. To test this, we simulated neutral sequences along the 36 species phylogeny and identified the 95% upper quantile of the GERP score distribution to use as a cutoff to identify mutations undergoing purifying selection. We then applied this cutoff to data simulated with different strengths of purifying selection. GERP has near-perfect power to correctly identify sites with moderately ($N_e s$<-2) to strongly deleterious mutations (orange curve in Fig 1B). Power drops dramatically for sites where mutations are nearly neutral.

Though GERP scores are often applied to coding regions of genes, they are calculated without using or modelling the codon structure of protein-coding genes. To examine how GERP scores behave under a codon-based evolutionary model, we simulated sequences under a codon model where mutations either change the coding amino acid (nonsynonymous) or do

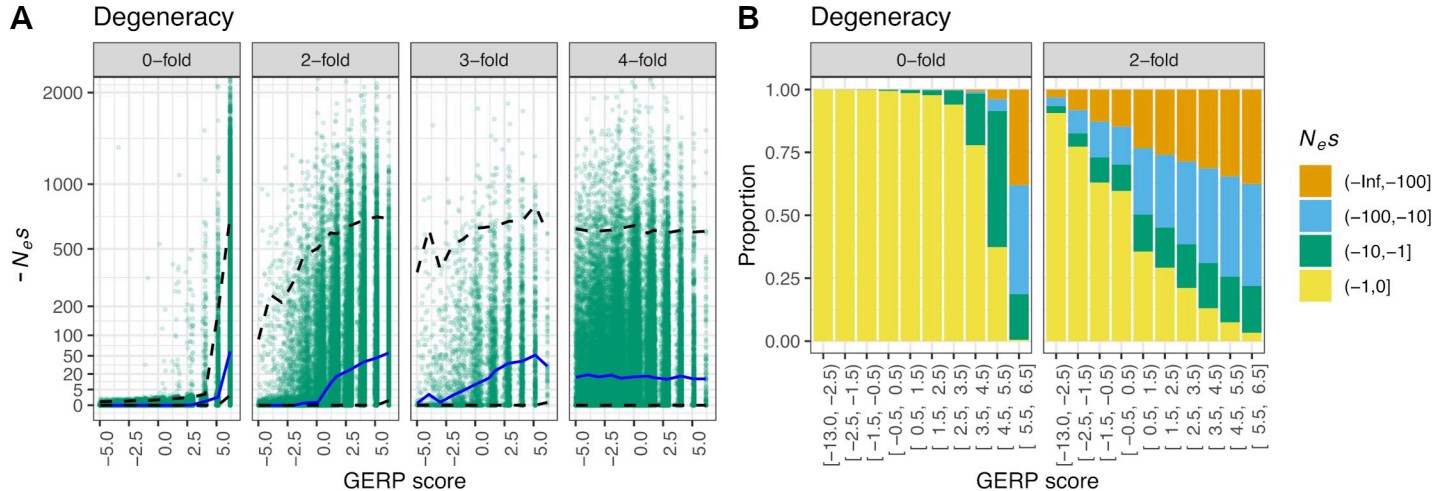

**Fig 2. GERP scores and $N_e s$ values under a codon-based model of evolution.** (A) $N_e s$ values of nonsynonymous mutations as a function of GERP scores for different degrees of codon degeneracy. Codon degeneracy and $N_e s$ value are observed in humans, whereas simulations are run across the entire 36 species tree. Note that a nonfunctional site in humans is considered to have a $N_e s$ value of zero, whereas a 4-fold degenerate site can have $N_e s$ values different from zero. The blue line represents the median $N_e s$ value given a specific GERP score, whereas the dashed lines represent the 2.5% and 97.5% quantiles. (B) Distribution of $N_e s$ values for GERP scores at 0-fold and 2-fold sites. Note that the $N_e s$ values are distributed differently for the same GERP scores depending on the type of site.

not change the amino acid (synonymous; see Methods). We simulate codons as a sequence of three nucleotides, and for each simulated codon, the selection coefficient of an amino acid changing mutation was assigned a value sampled from a mixture of a gamma distribution and a point mass at neutrality as inferred in Kim et al. [44]. Synonymous mutations were assumed to be neutral. For 0-fold degenerate sites, i.e. sites where every possible mutation is nonsynonymous, we see that GERP has reasonable power to distinguish nearly neutral mutations from more strongly deleterious ones (Fig 2A). For example, GERP scores <0 are almost only seen for sites that have $N_e s$ > -1, whereas GERP scores >5.5 are strongly associated with purifying selection ($N_e s$<<-1). However, only about 67% of GERP scores between 4.5 and 5.5 are subjected to purifying selection (Fig 2B). GERP scores are less predictive for amino acid changes at 2-fold and 3-fold degenerate sites than at 0-fold degenerate sites (Fig 2; S1 Fig). At 3-fold degenerate sites, as much as 21% of the sites with $N_e s$ < -1 show GERP scores that are <0 as a result of neutral changes occurring at these sites throughout the evolutionary history of the phylogeny. Mutations at 4-fold degenerate sites are neutrally evolving in these simulations. However, due to the long evolutionary time span covered by the 36 mammalian species phylogeny, in codon models, sites that are currently 4-fold degenerate in a certain species could have been 0, 2, or 3 fold degenerate sites at previous times and could have experienced deleterious amino-acid changing mutations during their history. Nonetheless, for these sites, we see in our simulations that the average strength of selection is unrelated to the GERP score (Spearman's $\rho$ = 0.0061, p = 0.16).

In sum, due to the redundancy of the genetic code, the same GERP score can be associated with different $N_e s$ values, depending on where it occurs in the codon, complicating the mapping of GERP scores to selection coefficients.

### Functional turnover at noncoding sites

Thus far our models have assumed that the selection coefficient at a given site has remained constant over evolutionary time. However, there is some evidence of rapid functional turnover of noncoding constrained sites [25,30,33,37,38]. Further, even if the regulatory architecture

has remained constant over evolutionary time (i.e. the same amount of gene product is needed and the same transcription factors are binding in the same general area of sequence), the selection coefficients can change over time due to changes on the genetic background on which mutations arise. Such turnover implies that mutations at a site may be under selection during some time points, while neutrally evolving in other species or at other points in time. To investigate the performance of the GERP score under scenarios with evolutionary turnover, we simulated data over the 36 species phylogeny using a model where a site transitions between functional and non-functional according to a Markov process (see Methods; S2 Fig). The rate of turnover from functional to non-functional was set to 2.48 turnover events per neutral substitution, as estimated for noncoding elements in Rands et al. [25]. The rate from non-functional to functional was set to 0.19, assuming an equilibrium of ~ 7% functional sequence. The selection coefficients of mutations at functional sites are assumed to be distributed according to Torgerson et al. [45], whereas mutations at non-functional sites are assumed to be neutral.

When there is no functional turnover, the distribution of GERP scores is reasonably predictive of the strength of selection acting on a site (see above; Fig 3A). Specifically, 82% of the sites that show a large GERP score (>5.5) are selected with $N_e s < -1$. Sites where mutations evolve neutrally have smaller GERP scores. Assuming functional turnover as outlined above (i.e., according to the estimates in [25]) results in a different pattern. With functional turnover, we observe sites that are under strong selection within the human lineage but have very small and even negative GERP scores (Fig 3A). Turnover of functional sequence also results in neutrally evolving sites within the human lineage showing large GERP scores (Fig 3B). For example, approximately 61.6% of GERP scores >5.5 in our simulations are from sites that are not functional in humans, i.e. mutations segregating at those sites in humans would be neutral, but the GERP score at those sites would strongly signify selection. For comparison, in a model without turnover, only 15.9% of mutations at sites with a GERP score >5.5 are neutrally evolving in humans. Less extreme GERP score cutoffs have a larger proportion of neutral sites even under the model without functional turnover (e.g., 76.0% neutral sites for GERP score >4), which

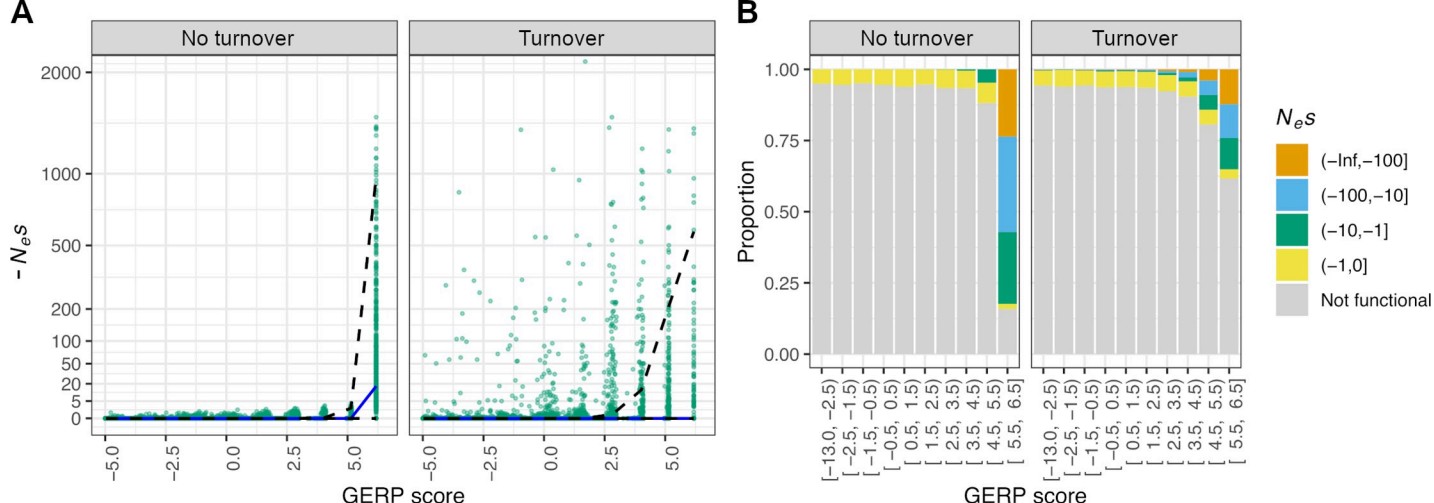

**Fig 3. Turnover of selected sequence disrupts the relationship between GERP scores and $N_e s$ values.** (A) $N_e s$ values as a function of GERP scores for a model without turnover of functional sequence across the 36 species tree (left) or where there is turnover modelled according to our Markov model (right). The turnover rate is estimated in Rands et al. [25] for noncoding elements. Green dots denote selected sites and yellow dots denote neutral sites, as observed in humans. The blue line represents the median $N_e s$ value given a specific GERP score, whereas the dashed lines represent the 2.5% and 97.5% quantiles. (B) Distribution of $N_e s$ values for GERP scores when there is no turnover (left) and when there is turnover of functional sequence (right). Note that when there is turnover, the majority of the sites with high GERP scores (>5.5) are not functional.

only worsens under the turnover model (85.1%). Thus, even a small amount of functional turnover can dramatically limit the utility of GERP scores at detecting mutations under selection.

Finally, we explored the power of GERP scores to detect purifying selection that only occurs within a specific subset of the phylogeny. Mutations at these sites are neutrally evolving on the remaining parts of the tree. As done previously, the entire 36 species were used to compute GERP scores. When considering purifying selection only on the 18 supraprimates (e.g. rodents, lagomorphs, treeshrews, colugos, and primates), power reaches a maximum of about 23% (Fig 1B). When considering purifying selection restricted to either the primates or just the human lineage, GERP has essentially no power to detect purifying selection. In sum, consistent with our simulations under the Markov model of turnover, the general application of GERP scores will have little power to identify lineage-specific purifying selection. We explore the sensitivity of our conclusions to various modeling assumptions (S1 Text, S4 Fig–S7 Fig).

## Optimal tree size for GERP scores

The simulations thus far have used a phylogenetic tree of 36 mammalian species. We next investigate the effect of the tree size, i.e. the sum of branch lengths of the phylogenetic tree, on the power to detect sites under selection. There are several reasons why the tree size may vary. First, the tree size varies across the genome due to missing data in the multi-species sequence alignment, leading to branches that have to be excluded due to missing data (S3 Fig). Second, there is interest in expanding the number of genomes sequenced [28,29], increasing the number of branches of the tree. The impact that larger phylogenetic trees will have on the utility of GERP has yet to be explored. Lastly, we saw that with sequence turnover, or selection on specific branches of the phylogeny, using the entire tree to compute the GERP scores was drastically limited in power. It may be possible to use subsets of the tree to improve power.

To investigate the power of GERP to detect sequence under purifying selection, we generated trees with a wide range of tree sizes. For this analysis, we used a 100 vertebrate tree downloaded from the UCSC genome browser. Starting with the two branches from the tree that connect humans and chimps, we successively added species to increase the size of the tree in the smallest possible increments of total branch lengths, until we reached a full tree of 100 vertebrate species, with a total tree size of 18.5 expected substitutions per site. Then, we simulated alignment data conditional on the respective trees under neutrality and with varying levels of selection (see Methods). When there is no sequence turnover, increasing the tree size results in an increase in statistical power to detect selected sites (Fig 4). The increase in power is most dramatic for weak selection ($N_e s >$-2). For strong purifying selection ($N_e s <$-100), there is close to 100% power even for shallow tree depths of 4 expected substitutions.

We next investigated scenarios with turnover of selected sequence. We evaluated a level of turnover as estimated for noncoding elements [25] and an intermediate level with a rate of turnover half of the estimated value. In both cases, the power to detect functional sequence in humans decreased dramatically relative to the no turnover scenario, consistent with the simulations described above (Fig 4). Further, there is greater power to detect stronger selection. However, we now find that increasing the phylogenetic tree size does not monotonically increase statistical power. There is no improvement in statistical power with increasing tree size for very weak selection ($N_e s <$ -0.25). Moreover, for strong selection, power peaks at a tree size of about 6.5 substitutions and, after that, decreases with increasing tree size. The reason for this decrease in power is that adding more species at this point leads to a decoupling of the functional status in humans (or any other focal species) and the functional status in the other species and therefore level of conservation across the phylogeny. In other words, mutations at

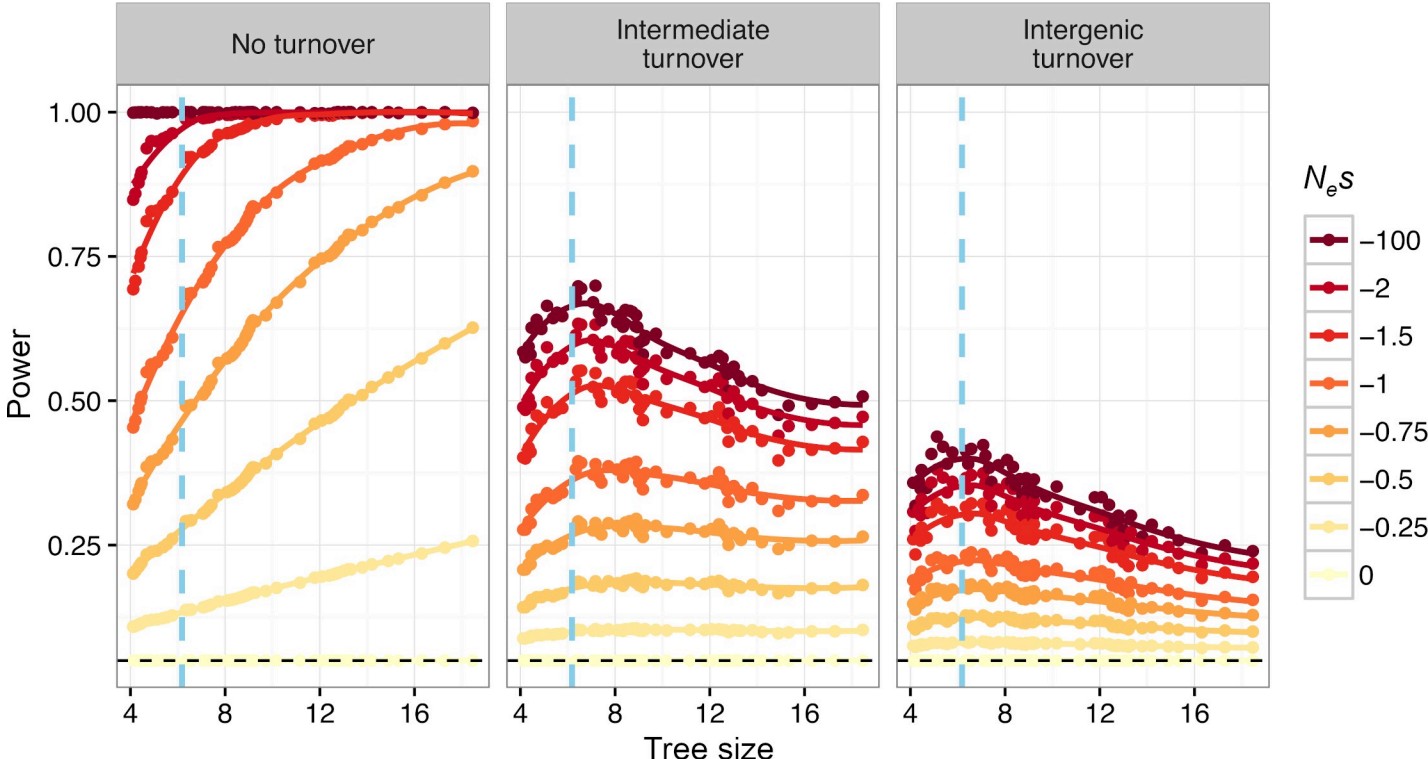

**Fig 4. Power to detect purifying selection using GERP scores as a function of tree size.** Colored lines denote different strengths of purifying selection. Tree size is defined as the sum of lengths of all branches of the tree. Branch length is measured as expected neutral substitutions, i.e. a branch with length one has on average one neutral substitution. The tree size is varied by including/excluding species from a phylogenetic tree of 100 vertebrates (see main text). Left panel shows no turnover. Right panel shows intergenic levels of turnover with turnover rate as estimated in Rands et al. [25] for noncoding elements. Middle panel shows intermediate turnover with a rate half of that in the right panel. Blue vertical lines denote the tree size of the 36 mammalian species tree that is commonly used for calculation of GERP. See S2 Text for further discussion of alternate strategies for adding species.

highly constrained sites that are functional across large parts of the phylogenetic tree might have switched to non-functionality in humans, and vice-versa. When considering high levels of turnover, like those that may be expected in intergenic regions, this de-coupling becomes stronger and statistical power decreases even more (Fig 4). These results suggest that optimal tree size depends on the strength of purifying selection as well as whether there is turnover of functional sequence. Adding additional sequences does not always improve power and in fact can decrease power for identifying strongly selected elements with high levels of turnover.

## Comparing evolutionary turnover models using data

The distribution of GERP scores across the human genome is a summary statistic of sequence conservation. Here we leveraged this summary statistic to compare different models of turnover of selected sequence to each other. We do this by fitting evolutionary models to the empirical GERP score distribution computed from commonly used sequence alignment of data from 35 mammalian species to the human genome [3,5,19,41,43]. Because the number of species for which there is data in the multi-sequence alignment varies across the genome, we partitioned the genome into multiple bins. Each bin represents sites with one particular tree size (S3 Fig). Overall, a total of 268 bins were considered, with tree sizes ranging from 3.5 neutral substitutions to 6.18 neutral substitutions per site. Fig 5 shows examples of the GERP score distribution for sites corresponding to small, intermediate, and large tree size (3.52, 4.69, and

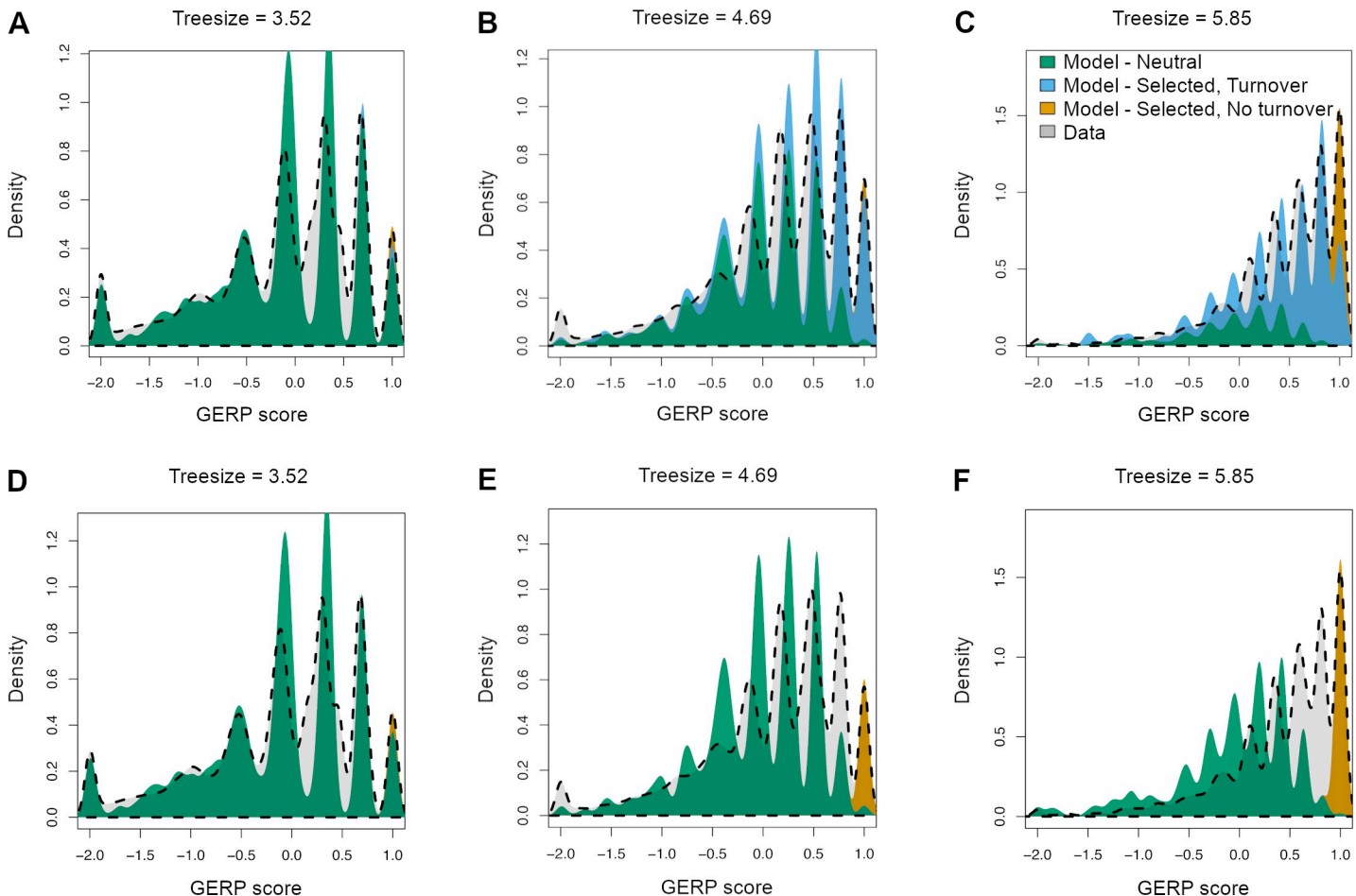

**Fig 5. Fit of models of purifying selection to the empirical GERP score distribution for different tree depths.** Dashed gray lines indicate the empirical distribution of GERP scores. The 3 plots in each row denote the distributions for different depths in the multi-species sequence alignment. The GERP scores were normalized by dividing each score by the largest possible score given the tree size (see Methods). (A-C) Fit of a model with 3 categories of sites: neutral, selected, and turnover (see text). (D-F) Fit of a model with 2 categories of sites: neutral and selected. Note that the model with turnover provides a more satisfactory fit to the empirical data.

5.85, respectively). As expected, the GERP score distribution differs across these three tree sizes. Sites with the largest tree sizes have a higher proportion of sites with the largest possible GERP score. This result is unsurprising since high sequence conservation across species is likely to improve the quality of the multi-species alignment, leading to greater tree sizes.

We next fit a mixture distribution to the empirical GERP score distributions. The mixture distribution consisted of three categories of simulated mutations. The first category consisted of sites where mutations are neutrally evolving across the entire phylogenetic tree (category N). The second category consisted of sites where mutations are consistently under purifying selection across the entire tree, such that substitutions would not occur and GERP scores would show the maximum value (category C). The third category consisted of sites that had experienced functional turnover or had changed selection coefficients over the timescale of mammalian evolution (category TO). That is to say, at these sites mutations are deleterious in the human lineage but could have been neutral on other branches of the phylogeny. Here we used a turnover model based on the turnover rate of noncoding elements from Rands et al. [25]. When the site was under selection, no substitution would occur. However, when the site was neutral, substitutions could occur at the neutral rate (see Methods). The turnover model

allows substitutions to occur on branches that are non-functional but not along branches that are functional. The parameters of the mixture distributions estimated from the data are the proportions of sites in each of the three categories (see Methods). This mixture distribution can fit the empirical distribution of the GERP scores relatively well for all three tree sizes (Fig 5A–5C). The smaller GERP scores indicate less sequence conservation and such sites are mostly fit by the neutral component. The very high GERP scores, indicating a high degree of sequence conservation, are best fit by sites in category C (orange in Fig 5C). The turnover model can explain the intermediate GERP scores that are neither explained by sites in category N or C (blue in Fig 5B and 5C).

We next assessed how well a model fits the data by examining the overlap of the distribution of standardized GERP scores observed in the data with the distribution of standardized GERP scores under a certain model ($O_{Model}$; S8 Fig). $O_{Model}$ was measured from kernel density estimation of both distributions (see Methods for more details). A value of $O_{Model}$ of zero indicates no overlap of the two distributions, a value of one indicates perfect overlap, a value between zero and one indicates intermediate overlap. The parameters for each mixture model (i.e., the mixing proportions of sites in category N, C, and TO) were chosen such that they maximize $O_{Model}$, which means that the model fits optimally to the data.

The estimated proportion of sites in the neutral, selected, and turnover categories varies as a function of tree size (Fig 6A). The shorter tree depths have a more neutral sequence, as expected, considering that neutrally evolving sequence contains more substitutions and is thus harder to align. Importantly, more than half of the sequence with large tree sizes is accounted for by the turnover model. The remainder is split equally between the N and C categories. While the fit of the three-component mixture model is visually satisfying, we next compared the fit of a model that did not include evolutionary turnover (Fig 6B). This mixture model only contains the two components N and C. Visually, this two-component mixture model does not fit the empirical distribution of GERP scores very well for the medium and large tree sizes (Fig 5E and 5F). Specifically, the model without turnover cannot account for sites with intermediate GERP scores (light gray shading in Fig 5E and 5F).

To more formally compare the fit of the different models, we simulated sites under two- and three-component mixture models (N+C, N+TO, N+TO+C) as well as sites under the purely neutral model (N). We ran 500 replicates, and for each replicate, we mimicked the number of sites that are observed in the empirical data for each tree size (S3 Fig). The parameters for the simulations are from the tree-size specific estimates that we derived from the 36 species alignment data (Fig 6A and 6B). First, we test the performance of our method for estimating the proportions of the three components N, C, and TO. We find that our method leads to unbiased and accurate estimates of the proportion of sites from each of the three components, except for the highest tree sizes >6 where the small number of available sites leads to larger uncertainty in the estimates (S9 and S10 Figs).

Next, we developed a test statistic ($\Lambda$) for model comparison, i.e. for evaluating if a more complex model fits significantly better to the data than a less complex model. The test statistic $\Lambda$ is derived by computing the log-ratio of one minus the maximized overlaps $O_{Model1}$ and $O_{Model2}$ as $\Lambda_{Model1 \; vs \; Model2} = 2 \, (log(1-O_{Model1}) - log(1-O_{Model2}))$, where $Model1$ refers to the null model with fewer parameters. A large value of $\Lambda_{Model1 \; vs \; Model2}$ indicates that $Model2$ fits better to the data than $Model1$. We derive two null distributions for this test statistic: one using simulations under a model with only neutral sites (N; Fig 7A), and one using simulations under a model that contains neutral and selected sites, but no turnover (N+C; Fig 7B). Further, we calculate four versions of $\Lambda$, each one comparing two types of models (N vs. N+TO; N vs. N+C; N vs. N+TO+C, and N+C vs. N+TO+C).

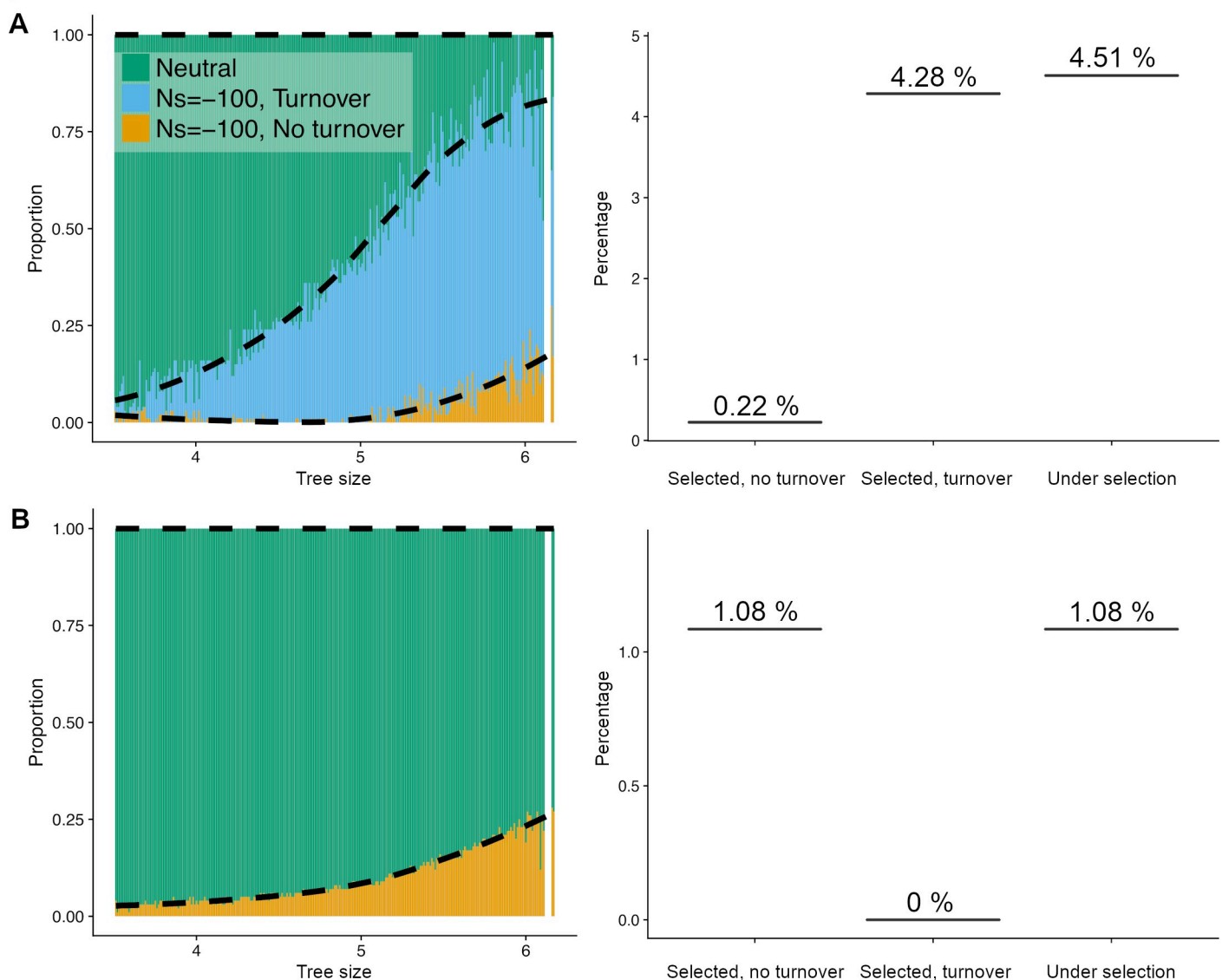

**Fig 6. Amount of the noncoding human genome under purifying selection for different models.** Left panels show the proportion of sites falling in each category of the mixture component as a function of tree size. Right panels show the proportion of the genome falling in the selected categories under the model. (A) The full model including sequence turnover (N+C+TO) that better fits the data. (B) A model without sequence turnover (N+C).

First, we simulated 500 test datasets under the N model to derive null distributions of $\Lambda_{\text{N vs. N+TO}}$, $\Lambda_{\text{N vs. N+C}}$, and $\Lambda_{\text{N vs. N+TO+C}}$. Next, we tested where the empirically observed values of $\Lambda_{\text{N vs. N+TO}}$, $\Lambda_{\text{N vs. N+C}}$, and $\Lambda_{\text{N vs. N+TO+C}}$ fell in the relevant null distributions. In all three cases, the observed test statistic falls well outside the range of the 500 simulated values (Fig 7A), suggesting that any of the three mixture models fit better to the data than the null model of neutral evolution (p < 0.01). Second, we simulated 500 test datasets under the N+C model and examined where the observed value of $\Lambda_{\text{N+C vs. N+TO+C}}$ fell relative to this simulated null distribution. We again find that the observed test statistic falls well outside the range of the 500 simulated values (Fig 7B), suggesting that adding the turnover component to the model significantly improves the fit (p < 0.01).

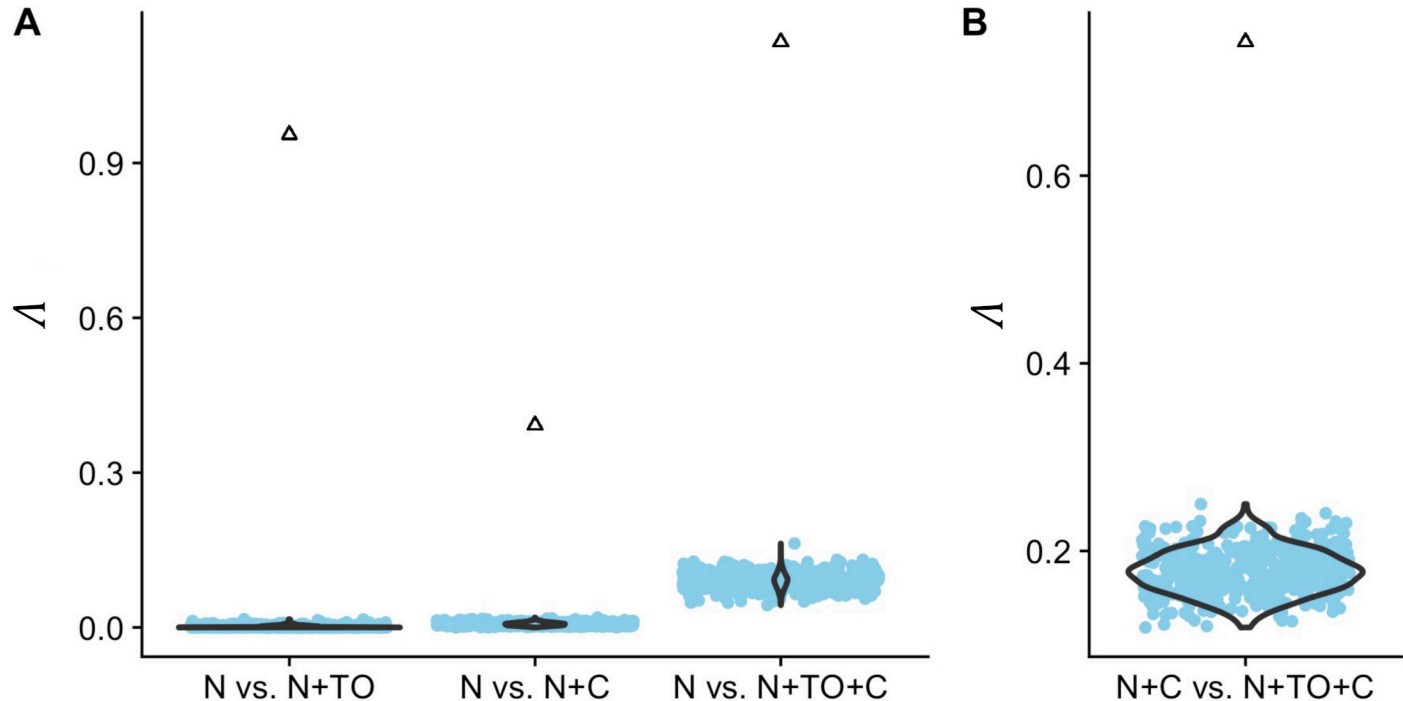

**Fig 7. The observed values of Λ fall outside the null distributions.** The null distribution (blue points) is derived from 500 simulations under the respective null model, i.e. assuming only neutral sites in (A) and neutral plus constantly selected sites in (B). The triangles denote the empirically observed statistics. In all cases, the null hypothesis is rejected with p<0.01.

Finally, we repeated the same testing procedure for each tree size separately and calculated p-values as the proportion of the 500 simulations that have a larger value of Λ than that of the data. We find that for tree sizes larger than approximately 5, the full model (N+C+TO) fits significantly better than both the N model and the N+C model (p < 0.05; S11 Fig). Below a tree size of 5, adding a selected component (N+C or N+TO) seems to significantly improve the fit over the neutral model (N). However, the full model (N+C+TO) does not improve over the N +C model, the N+TO model, or the neutral model (p > 0.05; S11 Fig). This is in agreement with the empirically estimated mixture proportions (Fig 5A), where the proportion of the C component approaches zero below tree size 5 and only the N and TO components remain.

In sum, we find that if the empirical data truly evolved under a scenario where mutations are either neutrally evolving or have the same selective effect across the phylogeny, we would be unlikely to see the observed improvement in fit by adding the turnover component to the N +C model. Taken together, these results imply that intergenic regions consist of a mix of sites: sites where mutations are neutral, sites where mutations are consistently selected across the full phylogeny, and sites where mutations are selected only in sub-parts of the phylogenetic tree.

## Inference of the amount of the human genome under purifying selection

Finally, summing over the different tree sizes weighted by their relative proportions across the genome, we can estimate the proportion of sites genome-wide in each category. We restrict our estimate to noncoding regions that show high mappability and remove repeat-rich regions. When considering the full (N+C+TO) model, 4.51% of the noncoding genome is inferred to be under purifying selection, with the majority of the 4.51% coming from the

**Table 1. Performance of GERP scores to identify noncoding sites containing deleterious mutations in the human lineage.**

| Threshold | % of sites > threshold that are neutral (FDR) | % of selected sites > threshold (power) | % of genome filtered | % of selected sites not filtered |
|---|---|---|---|---|
| >4 | 9.1% | 30.1% | 1.5% | 69.9% |
| >3 | 23.7% | 54.4% | 3.2% | 45.6% |
| >2 | 41.8% | 71.3% | 5.5% | 28.7% |
| >1 | 56.1% | 82.2% | 8.5% | 17.8% |
| >0 | 64.8% | 89.7% | 11.5% | 10.3% |

turnover category (Fig 6A). The model without turnover predicts a greater proportion of sites in the C category (1.08%), however, the overall proportion of the genome under purifying selection is substantially smaller than under the model with turnover. Given that the model with turnover fits the data significantly better than the model without turnover, we estimate that at least 4.5% of the noncoding human genome has been impacted by purifying selection along the human lineage. Adding about 1% functional coding sequences [46] to this proportion would make the total proportion of the genome under selection to be close to 5.5%.

Our estimated model of sequence evolution also allows us to test the effectiveness of GERP scores to prioritize or filter selected mutations at noncoding sites. GERP scores are used to identify sites where mutations may have an impact on fitness. For example, variants with a GERP score >4 are often assumed to be deleterious [3,4]. While this cutoff has a low false discovery rate (FDR; i.e. the proportion of sites with GERP>4 where mutations are not under selection) of 9%, the power to detect selected mutations is seriously limited (Table 1, S12 Fig). For example, only 30% of selected sites would be identified by this approach. Put another way, mutations in approximately 80.8Mb of the noncoding human genome are predicted to have an effect on fitness in humans, but would not show extreme GERP scores. Using less stringent GERP score thresholds to identify putatively deleterious mutations will increase the power, but with an increase in FDR (Table 1, S12 Fig).

GERP scores are also used to remove sites where mutations may be affected by purifying selection for demographic inference that assumes mutations are neutrally evolving. Retaining only sites with a GERP score <2 would remove ~5.5% of sites in the noncoding genome (Table 1, S12 Fig). However, under our full (N+C+TO) model, this cutoff would miss ~29% (33.2Mb) of noncoding sites in which mutations would be deleterious. Removing more of the genome can help mitigate this effect, but even in the extreme case of removing ~11% of the genome with GERP scores >0, this still leaves 10% of the selected sites behind.

In sum, filters based on GERP scores might not be effective for applications that rely on identifying selected sites with high sensitivity and specificity.

## Discussion

Here we use population genetic models over deep time scales to help interpret patterns of GERP scores seen for the human genome. We find that GERP scores are immensely useful for distinguishing mutations under purifying selection from mutations that are neutrally evolving if these evolutionary forces have not changed throughout deep evolution. Real-world complications, like codon models of evolution, missing data, and turnover of functional sequence can greatly limit the utility of GERP scores to identify sites where mutations are under selection. Lastly, a model including functional turnover fits the genome-wide distribution of GERP scores significantly better than a model where mutations at sites are either under selection or neutrally evolving across the entire mammalian phylogeny. This best-fitting model suggests that mutations in about 4.51% of noncoding sites are under purifying selection in the human lineage.

Our estimate that 4.51% of noncoding sites in the human genome experience deleterious mutations is in line with previous estimates based on conservation patterns [33]. However, it is most likely an underestimate of the fraction of functional sequence, for several reasons. First, our analysis does not detect functional sequences that are evolving very rapidly and/or are subjected to positive selection [47]. Positive selection increases divergence above neutral levels and thus would lead to negative GERP scores that are interpreted as neutral in our approach. Second, GERP scores are based on a neutral reference tree with branch lengths estimated from four-fold degenerate sites [19]. However, there are indications that synonymous sites of vertebrate genomes are also subject to purifying selection [48]. For example, it was shown that the overall divergence between chimpanzees and humans is 39% lower at four-fold degenerate sites than at intergenic sites [49]. Thus, the rate of evolution at four-fold degenerate sites is likely an underestimate of the rate of neutral evolution at intergenic sites. Using four-fold degenerate sites as neutral reference is, therefore, a conservative approach as it biases the neutral GERP score distribution to negative values and purifying selection has to be strong enough to overcome this bias. Finally, our estimate of the fraction of sites under purifying selection does not measure selection against insertions or deletions (indels). For example, indels may induce frameshifts in coding regions or secondary structure changes in RNAs, suggesting that stronger purifying selection may often act upon them than on nucleotide changes in the same region. This might explain the discrepancy between our estimates of the fraction of functional sequence and a recent estimate based on indels that suggests that about 7% of the noncoding human genome is subject to purifying selection [25].

Our finding that the majority of the noncoding sequence under purifying selection in the human genome has not been under purifying selection across the entire mammalian phylogeny does not necessarily mean that the regulatory architecture or selective pressures themselves have changed over time. The overall amount of stabilizing selection on gene expression could have remained constant over time. Rather, selection coefficients at particular sites could change over time due to the genomic background on which a given mutation occurred [40]. For example, a neutral substitution could fix in a regulatory region, making a mutation at another site be less deleterious than originally found. Thus, changes in selection coefficients across the phylogeny at particular sites are expected simply due to the stochastic nature of the evolutionary process governing which other neutral and nearly neutral mutations have become fixed in particular lineages.

While the turnover component was required in the mixture model to explain the intermediate GERP scores across the human genome, it is in principle possible that very slightly deleterious mutations also could account for some of these intermediate scores. However, we ignore such slightly deleterious mutations since they are relatively rare and most selected mutations either have a selection coefficient such that they do not fix ($N_e s < $ -3) or fix at an effectively neutral rate (-0.1 $< N_e s < $ 0). This is supported by simulations assuming a distribution of selection coefficients of intergenic elements [45]. Under this distribution, the vast majority of mutations either have an expected substitution rate of close to zero, or a neutral substitution rate (S4 Fig). For the same reason, we find that varying effective population size and thus varying effectiveness of selection across the phylogenetic tree does not substantially alter the relationship between $N_e s$ and the GERP score (S1 Text and S7 Fig). Thus, intermediate GERP scores are most likely explained by functional turnover and not by mutations with only slightly deleterious selection coefficients or varying effective population size across species.

While comparative genomic studies have had some success at identifying functional sites [17–19,50–55], our present work shows that improvements can be made. First, one improvement would be to use alternate measures of conservation in coding regions of genes [15,16], or by combining existing codon models of evolution with the GERP framework. Second, because

GERP is less effective when there is functional turnover, and at distinguishing weak selection from strong selection, improvement can be made by using polymorphism-based measures of purifying selection [18,23,56–58]. These methods should only improve as the sample sizes of human genomes become greater [59]. Lastly, methods that seek to combine comparative genomic information with functional data [17,18,26,60,61] may also offer improved power to identify functionally important sites.

For evolutionary genetic studies, our work suggests that GERP may be a reasonable approach to identify which coding sites, if mutated, would give rise to a deleterious mutation. Because coding sites are thought to not undergo much functional turnover, GERP should still have some utility. However, the common approach [3–5] of assuming that the GERP score is proportional to the deleteriousness of a given mutation does not appear to work well. The prospects of using GERP to compute genetic load at noncoding mutations is likely to be even more tenuous, due to functional turnover (Table 1). For example, we estimate that mutations in ~115 Mb of the noncoding genome are under purifying selection in the human lineage, but only 30% of these sites would have a GERP score >4. Using GERP scores to filter sites where mutations may be deleterious will also miss a substantial proportion of sites where mutations are under selection. For example, filtering the ~6% of the genome with GERP scores >2 will miss 29% of the selected sites. The impact of these effects on downstream analyses remains to be quantified in future work.

Our work has implications for future comparative genomic studies. The finding that increasing the number of species used in the multi-species alignment does not always result in an improvement in power suggests the need to carefully assess the utility of sequencing additional genomes for conservation-based measures of selection. Another challenge to adding sequence to compute GERP scores is the need to generate accurate multi-species sequence alignments. Because all of our simulations assumed no errors in multi-species sequence alignments, performance is likely to be worse with alignment errors. From this point of view, computing conservation scores from closely related species with a shallow phylogenetic relationship is advantageous since the genomes have a highly correlated functional state and are readily alignable to the focal species. However, if the overall tree size is too small, then conservation (i.e., a lack of substitutions) is harder to detect and power is low. This leads to a tradeoff between tree size and relatedness between the included species (see also S2 Text). Simulations under our turnover model can be used to optimize the choice of species that are included in a conservation score analysis, given a specific turnover rate and selection strength (Fig 4; S2 Text). As projects like Genome 10K [28,29] continue to develop, there will be sequence data from a cornucopia of species. There is a need for conservation metrics to be computed on different phylogenetic scopes, with different tree sizes, to optimize power for the turnover rate and selection strengths of specific functional elements.

The amount of the human genome under the direct effects of purifying selection has remained controversial. Previous estimates from comparative genomic approaches put a lower bound around 3–4% [33]. However, these studies require a class of putative sites from which to estimate the neutral substitution process. Other studies, relying on indels, found that the percentage of the genome under purifying selection is likely to be slightly higher. Our current approach estimates that at least 4.5% of the noncoding human genome has been subjected to purifying selection. As opposed to previous approaches, we explicitly model functional turnover and control for the relation between the level of conservation and the amount of missing data. Thus, our approach provides an independent line of evidence that about 4.5% of the noncoding genome is under purifying selection. Importantly, we directly compare the fit of models with and without turnover to the genome-wide GERP score distribution. Because our statistical approach makes few assumptions (see S1 Text for a detailed discussion of the assumptions

behind our approach), it provides robust statistical support for the turnover of selected sequence. Importantly, our approach suggests that many of the noncoding sites under purifying selection fall in the turnover category. As such, our work argues that evolutionarily important sequences have changed over millions of years of evolution.

## Methods

### Simulating deleterious substitutions along a phylogenetic tree

A neutral tree of 46 vertebrate species was downloaded from the UCSC genome browser (http://hgdownload.cse.ucsc.edu/goldenPath/hg19/multiz46way/46way.nh). A subset of species was then selected to arrive at the tree of 36 mammalian species that is commonly used for GERP analysis [3,5,19,41,43]. The length of a branch reflects the neutral substitution rate along the branch, i.e. the expected number of neutral substitutions along this branch per site. Simulations of alignment data were done with the software *pyvolve* [62], under the *nucleotide/HKY85* model and setting the transition to transversion ratio to 2. The equilibrium frequencies were assumed to be equal for all four nucleotides. To test robustness of the GERP score distribution, a range of different nucleotide evolution models were simulated as well (see S1 Text). Deleterious mutations have a reduced substitution rate as a function of the strength of selection according to Eq 1. We simulate deleterious mutations by scaling the full tree by a factor ω which takes into account the reduced substitution rate as a result of purifying selection.

### Estimating the rate of substitutions with GERP++

We used the program *gerpcol* from the GERP++ software [19] to estimate the number of "rejected substitutions" (RS), or GERP score. The GERP score can be viewed as the number of substitutions "rejected" by evolutionary constraint. *Gerpcol* estimates nucleotide frequencies from the alignment data to use in the calculation of the GERP score, as was done for the robustness analyses in S1 Text and S13 Fig. Generally, we can estimate these frequencies accurately from whole-genome data, so we removed this source of estimation variability and fixed the nucleotide frequencies to the known simulation values (i.e. equal proportions of nucleotides). This was done by changing the source code of *gerpcol*.

### Simulations under the codon model

The codon simulations were done with *pyvolve* under the *codon* model, again with a transition to transversion ratio of 2. The 'neutral_scaling' parameter was set to True to make sure that the tree reflects the synonymous substitution rate and not the overall substitution rate (see *pyvolve* manual). The codon model assumes a GY-style substitution matrix. The dN/dS ratio is set with the *omega* parameter in the *pyvolve* model specification. The ω parameter was calculated by using Eq 1, given a specific $N_e s$ value for nonsynonymous mutations. The nonsynonymous $N_e s$ values were sampled according to the best fitting-model in Kim et al. [44], where the $N_e s$ values of mutations are distributed according to a mixture of 82.6% gamma-distributed $N_e s$ values (shape = 0.343, scale = 334) and 17.4% neutral mutations.

### Simulations under the turnover model

We model functional turnover as a time-homogeneous random Markov process with two states, functional and nonfunctional, using the model proposed by Rands et al. [25]. This model makes the assumption that the rate of turnover is constant throughout time. Specifically, each simulated functional site will lose its function with a rate *b* and each nonfunctional site will gain function with rate *c*. The equilibrium proportion of functional sequence is thus *c/*

(*b+c*). As opposed to previous models that assume an infinite sites model [25], we consider the genome to consist of a finite number of sites and thus account for a reversion back to functionality of neutral but previously functional sequence.

We simulated functional turnover along a given phylogenetic tree. The functional state of the root branch was chosen randomly with probability c/(b+c) for being functional, and 1-c/(b+c) for being nonfunctional. The rates *b* and *c* are considered per one time unit of branch length, which is usually provided in units of expected neutral substitutions. The waiting time for a switch of the state along a branch was simulated as an exponentially distributed waiting time with rate parameter *c* for a branch that starts in a nonfunctional state, and *b* for a branch that starts in a functional state. Further, a branch can contain more than one switch along its length (e.g. from functional to nonfunctional, and back to functional), or no switch at all if the simulated waiting time is longer than the branch length. Whenever a branch splits into two sister branches along the phylogenetic tree, the functional state at the end of the parental branch is copied to the beginning of the two sister branches. The simulation of functional turnover is started at the root of the tree and then subsequently proceeds over all branches of the tree.

Simulation of alignment data was again done using the software *pyvolve*. However, the scaling of the simulated tree for deleterious sites was now done for each branch individually. Each branch length was scaled by a factor that is the sum of ω times the proportion of the branch that was in a functional state, and the proportion of the branch that was in a nonfunctional status. Thus, the proportion of the branch that is functional had a substitution rate that was reduced by the factor ω, whereas the remaining nonfunctional part had a neutral substitution rate.

## Estimating the proportion of sites under purifying selection

GERP scores were obtained from the University of California, Santa Cruz genome browser based on an alignment of 35 mammals to the human reference genome hg19. The allele represented in the human hg19 sequence was not included in the calculation of GERP scores. The GERP scores were further standardized by dividing by the expected number of neutral substitutions given the tree size. Thus, a value of zero indicates a neutral rate of substitutions, a value of one indicates no substitutions, and a negative value indicates a larger number of substitutions than expected under neutrality.

Next, a mixture model was fit to the observed GERP score distribution to provide a direct estimate of the amount of the noncoding genome under purifying selection, as well as the proportion of the genome where selection coefficients have changed over time. The mixture model includes three categories of mutations. The proportions of sites in each category are the parameters we estimated from the model. The first category consists of sites where mutations are neutrally evolving across the entire phylogenetic tree (category N). To generate the GERP scores under this model we simulated genetic data along the 36 species phylogeny using pyvolve and *gerpcol* as described above (*Simulating deleterious substitutions along a phylogenetic tree* and *Estimating the rate of substitutions with GERP++*). The second category consists of sites where mutations are consistently under purifying selection across the entire tree, such that substitutions would not occur and GERP scores would show the maximum value (category C). The third category consists of sites that had experienced functional turnover or had changed selection coefficients over the timescale of mammalian evolution (category TO). To generate GERP scores under this category, we used the turnover model of Rands et al. as described above (*Simulations under the turnover model*), with a rate parameter of turnover that was estimated from intergenic data [25].

Because GERP scores do not follow a common probability distribution [19], we used a nonparametric approach. We separately fit a kernel density to the empirical distribution of GERP

scores and as well as to the GERP scores simulated under a particular mixture model (see below). The density was estimated in R using the *density* function with a bandwidth of 0.05, a number of grid points of 5000, and a gaussian kernel. After calculating a kernel density of both empirical and simulated distribution, we then assessed the fit of different models using the overlap statistic ($O_{Model}$) of the distribution of standardized GERP scores observed in the data with the distribution of standardized GERP scores under the model (S8 Fig). $O_{Model}$ is calculated as two minus the sum of the absolute difference in density of model versus data at each of the 5000 grid points on the standardized GERP score axis, multiplied by 0.0006, the distance between two neighboring grid points. Thus, $O_{Model}$ is measuring the amount to which the two probability densities overlap. A value of $O_{Model}$ of zero indicates no overlap between the two distributions, a value of one indicates perfect overlap, a value between zero and one indicates intermediate overlap. The parameters for each mixture model (i.e., the mixing proportions of sites in category N, C, and TO) were chosen such that they maximize $O_{Model}$, i.e. such that the model fits optimally to the data. Maximization was achieved by an exhaustive grid search over a dense grid on the mixing proportions of the three components N, C, and TO, constrained on the proportions summing to one. The mixing proportions that lead to the largest overlap between model and data were defined as the estimates of the proportions. Simulations suggest that the estimates are accurate and unbiased (S9 and S10 Figs). Summing over the different tree sizes weighted by their relative proportions across the genome was used to estimate the genome-wide proportion of sites in each category. However, for this, we remove regions that are repeat-rich or show low mappability by filtering out challenging regions of the human genome using the bed file downloaded from https://github.com/Boyle-Lab/Blacklist/blob/master/lists/hg19-blacklist.v2.bed.gz. Most of these regions do not have any sequence alignment across species.

## Model comparison

To formally compare mixture models which include different numbers of mixture components (e.g. N+C vs. N+TO+C), we developed a test statistic ($\Lambda$) for evaluating if a more complex model fits significantly better to the data than a less complex model. $\Lambda$ is derived by computing the log-ratio of the maximized overlaps $O_{Model1}$ and $O_{Model2}$ as $\Lambda_{Model1 \ vs \ Model2} = 2(log(1-O_{Model1})-log(1-O_{Model2}))$, where *Model1* refers to the null model with fewer parameters. A large value of $\Lambda_{Model1 \ vs \ Model2}$ indicates that *Model2* fits better to the data than *Model1*, i.e. has a larger overlap statistic. We derive two null distributions for this test statistic, one using simulations under a model with only neutral sites (N; Fig 7A), and one using simulations under a model that contains neutral and selected sites, but no turnover (N+C; Fig 7B). Further, we look at four versions of $\Lambda$, each one comparing two types of models (N vs. N+TO; N vs. N+C; N vs. N+TO+C, and N+C vs. N+TO+C). We compute the four versions of $\Lambda$ from 500 simulations under the null models and then contrast this distribution with the $\Lambda$ statistic calculated from the data. The proportion of simulated $\Lambda$ statistics larger than the empirically observed value of $\Lambda$ is an estimate of the p-value. A confidence interval of the true p-value, i.e. the value that would be derived from an infinite number of simulations, can be computed by assuming that the number of simulated $\Lambda$ statistics that are more extreme than the observed value follows a binomial distribution, with the p-value as success probability and the number of simulations as number of trials. For example, if none of the 500 simulated statistics is more extreme than the observed value, then the 95% confidence interval of the true p-value can be computed with the R command *binom.test(x = 0, n = 500)* and is [0, 0.00735]. In this case, we would state that p<0.01.

## Supporting information

**S1 Text. Model assumptions and robustness of inference.**
(PDF)

**S2 Text. Optimally increasing power when adding species.**
(PDF)

**S1 Fig. Relationship between GERP scores and strength of selection at zero-fold codon sites.** The strength of selection, $N_e s$, is distributed according to a mixture distribution where 82.6% of mutations have gamma-distributed $N_e s$ values (shape = 0.343, scale = 334) and 17.4% of mutations are neutral. The green points represent selected mutations, the blue points neutral mutations. The red points are averages of $N_e s$ in bins of GERP scores. Note that the first codon position of four codons (CTC, CTT, CGT, CGC) shows a less consistent relationship between GERP score and strength of selection where even strongly selected sites can have GERP scores that are different from the maximum value. We thus classified these four classes of sites as two-fold instead of zero-fold.
(PDF)

**S2 Fig. Examples of simulated functional turnover on a tree of 100 vertebrates.** The rates of turnover from a functional state to a non-functional state along the tree for both coding and intergenic sites was taken from Rands et al. [25]. The back-mutation rate from a non-functional to a functional state assumes an equilibrium proportion of functional sequence of 7%. A red branch indicates functional state, a black branch indicates non-functional state. The simulations are conditioned on the human lineage (far left of the tree) being either in the functional state (columns one and three) or the non-functional state (columns 2 and 4).
(PDF)

**S3 Fig. Tree size distribution for the 36 mammalian species alignment.** The tree size is in units of expected neutral substitutions on the tree. It is taken from the first column of the GERP/gerpcol software output, which takes into account the reduced tree size due to missing data in some of the species.
(PDF)

**S4 Fig. Without turnover, most sites either show no substitutions or neutral levels of substitutions.** The distribution of expected substitutions (A) given a DFE for noncoding conserved elements from Torgerson et al. [45]. The assumed tree size of 5.85 should reflect almost perfect alignment in the commonly used 36 mammalian species alignment (same as in Fig 4C and 4D). Note that the vast majority of sites either experience an expected substitution rate of neutral sites (peak at 5.85) or zero substitutions. When assuming a Poisson distribution of substitutions on the tree to compute standardized GERP scores (B), the score distribution reflects a mixture of either strongly selected sites (peak at standardized GERP score of one) or a neutral distribution of substitutions (distribution centered at zero). However, the empirical distribution of standardized GERP scores (gray) contains a considerable density of sites with a score between 0.5 and 0.8 that cannot be fit by a model of a DFE estimated for noncoding conserved elements.
(PDF)

**S5 Fig. Estimation of the effective population size for 36 mammalian species.** (A) We collected values of synonymous diversity from the literature for a subset of 13 species and predicted synonymous diversity for the remaining 23 species by assuming a linear relationship between log(bodyweight) and log(diversity). (B) Synonymous genetic diversity is then

transformed into haploid effective population size by a simple linear interpolation, assuming a haploid population size of humans of 40,000 and of mouse of 1,160,000 (i.e. 20,000 and 580,000 individuals, respectively).
(PDF)

**S6 Fig. Phylogenetic tree of 36 mammalian species with estimated effective population sizes.** The effective population size of internal nodes was predicted assuming a Brownian model (see S1 Text).
(PDF)

**S7 Fig. Relationship between GERP scores and $N_es$ values in a phylogenetic model with varying effective population size.** Simulations assume the phylogeny and effective population sizes depicted in S6 Fig. (A) $N_es$ values as a function of GERP scores for a model without turnover of functional sequence across the 36 species tree (left) or where there is turnover modelled according to our Markov model, using the turnover rate from Rands et al. [25] for noncoding elements (right). The blue line represents the median $N_es$ value given a specific GERP score, whereas the dashed lines represent the 2.5% and 97.5% quantiles. (B) Distribution of $N_es$ values for GERP scores when there is no turnover (left) and when there is turnover of functional sequence. Note that when there is turnover, the majority of the sites with high GERP scores (>5.5) are not functional.
(PDF)

**S8 Fig. Definition of the overlap statistic $O_{Model}$.** The overlap $O_{Model}$ between the probability density of a statistic estimated from the data (blue area) and the probability density of the same statistic estimated under a model (red area) is defined as the area of overlap between the two distributions (striped area). Since probability densities integrate to one, the maximum value of $O_{Model}$ is one if both distributions fully overlap. If the two distributions do not overlap, the $O_{Model}$ is zero.
(PDF)

**S9 Fig. Estimation error for proportions of sites under different selection models.** The true minus estimated proportion of neutral sites (A), sites under functional turnover (B) and sites under constant selection (C) were computed using estimates from 500 replicated simulations. For each simulation, the parameters and tree sizes were chosen to reflect the empirical estimates and distribution (Fig 6A and S3 Fig). The gray shaded area reflects the mean +-2 SD. The blue line is a fitted loess curve to the data. It is centered on zero, suggesting that the estimates are unbiased across all tree sizes. Note however that the variance in error increases for tree sizes > 6 because of the small number of sites with alignment across most species.
(PDF)

**S10 Fig. Comparison of true and estimated mixture proportions under four different mixture models.** The estimated proportion of three different components of sites: neutral sites (N; green), sites under functional turnover (TO; blue) and sites under constant selection (C; orange), estimated from 500 replicated simulations. For each simulation, the parameters and tree sizes were chosen to reflect the empirical estimates assuming four different models: (A) only N sites, (B) mixture of N and C sites, (C) mixture of N and TO sites, and (D) mixture of N, TO, and C sites. The empirical (i.e. true) proportions are plotted as a black line. For most tree sizes the number of sites is large enough to reliably estimate the proportion of components under the four mixture models. However, the variance in error increases for tree sizes > 6 because of the small number of sites with alignment across most species.
(PDF)

**S11 Fig. P-values assessing the significance of improvement in model fit when adding additional components for each tree size.** The points denote p-values of rejecting the null hypothesis for each tree size. The p-value is calculated by comparing a null distribution of the test statistic $\Lambda$ with the value of $\Lambda$ observed in the data, where the test statistic $\Lambda$ is comparing the fit to the data of a more complicated model with more components with the fit of a simpler model with fewer components (see **Methods**). The null model and the alternative model are indicated at the top of each plot. The null distribution of $\Lambda$ was calculated from 500 simulations under the respective null model. The null model is assuming only neutral sites (A), neutral plus turnover sites (B, left column), or neutral plus constantly selected sites (B, right column). Points below the horizontal dashed line are significant assuming a false positive rate of 5%. The blue line is a smooth loess curve fitted to the points. Note that the data reject a null model of pure neutrality across all tree sizes (p < 0.05). For tree size > 5, the full model (N+C+TO) significantly improves the fit over the N+C model, suggesting a significant role of functional turnover for regions with large tree sizes, i.e. with alignment across most species. (PDF)

**S12 Fig. Effectiveness of using GERP scores to identify selected sites.** In our estimated full (N+C+TO) model, sites with tree size less than 3.5 are predicted to be exclusively neutral. Thus, we assume that these sites are filtered out accordingly. Blue denotes the number of sites in the human genome with GERP scores: (A) >4, (B)>3, (C) >2, (D) >1. Red denotes the number of sites inferred to be under selection in the human lineage. In C and D, the number of sites with GERP scores >2 or >1 is larger than the number of bases under selection. Note that any filtering strategy based on the GERP score can either have high sensitivity or high specificity, but not both. (PDF)

**S13 Fig. Robustness of the GERP score distribution to the nucleotide evolution model.** Violin plots of simulated GERP scores on a 36 species phylogeny assuming $N_e s$ values ranging from 0 to -8 in steps of 0.5. The colors indicate different models of nucleotide evolution. See S1 Text for details. (PDF)

**S14 Fig. Conceptual models for evaluating the increase in power per added species.** These trees in (A)—(C) represent a primary tree with different degrees of relatedness of a focal species to four other species. The scale of branch lengths in units of subs/site is shown in (A). To investigate the increase in power for detecting functional sites in the focal species, new species are added to this primary tree. The relatedness between the focal species and the added species is 0.4 subs/site (A), 2 subs/site (B), and 0.1 subs/site (C). In (D), every added species is closely related to species 2 (0.1 subs/site) but more distantly related to the focal species (0.4 subs/site). These trees are the basis for the simulations of alignment data and the computation of power for detecting functional sites in S15 Fig. See S2 Text for details. (PDF)

**S15 Fig. Evaluating the increase in power per added species for the conceptual models in S14 Fig.** Power is calculated based on the simulation of substitutions on the trees shown in S14 Fig., assuming different levels of turnover and selection coefficients. Left panels in (A)-(D) show no turnover. Right panels show intergenic levels of turnover with turnover rate as estimated in Rands et al. [25] for noncoding elements. Middle panels show intermediate turnover with a rate half of that in the right panels. See S2 Text for details. (PDF)

## Acknowledgments

We thank Eduardo Amorim, Chris Kyriazis, Jesse Garcia, and Jazlyn Mooney for comments on the manuscript as well as other members of the Lohmueller lab for helpful discussions. We thank Jessica Li for help with statistics for model comparison.

## Author Contributions

**Conceptualization:** Christian D. Huber, Bernard Y. Kim, Kirk E. Lohmueller.

**Data curation:** Christian D. Huber.

**Formal analysis:** Christian D. Huber, Bernard Y. Kim.

**Funding acquisition:** Kirk E. Lohmueller.

**Investigation:** Christian D. Huber, Bernard Y. Kim.

**Methodology:** Christian D. Huber, Bernard Y. Kim.

**Project administration:** Kirk E. Lohmueller.

**Resources:** Bernard Y. Kim, Kirk E. Lohmueller.

**Software:** Christian D. Huber.

**Supervision:** Kirk E. Lohmueller.

**Writing – original draft:** Christian D. Huber, Kirk E. Lohmueller.

**Writing – review & editing:** Christian D. Huber, Bernard Y. Kim, Kirk E. Lohmueller.

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
