## [Decision Letter · Decision Letter 0]

19 Mar 2020

Dear Dr Lohmueller,

Thank you very much for submitting your Research Article entitled 'Population genetic models of GERP scores suggest pervasive turnover of constrained sites across mammalian evolution' to PLOS Genetics. Your manuscript was fully evaluated at the editorial level and by independent peer reviewers. The reviewers appreciated the attention to an important problem, but raised some substantial concerns about the current manuscript. Based on the reviews, we will not be able to accept this version of the manuscript, but we would be willing to review again a much-revised version. We cannot, of course, promise publication at that time.

If you decide to revise the manuscript for further consideration at PLOS Genetics, please aim to resubmit within the next 60 days, unless it will take extra time to address the concerns of the reviewers, in which case we would appreciate an expected resubmission date by email to plosgenetics@plos.org.

[LINK]

We are sorry that we cannot be more positive about your manuscript at this stage. Please do not hesitate to contact us if you have any concerns or questions.

Yours sincerely,

Takashi Gojobori

Associate Editor

PLOS Genetics

Bret Payseur

Section Editor: Evolution

PLOS Genetics

Reviewer's Responses to Questions

**Comments to the Authors:**

Reviewer #1: GERP conservation scores are often used to evaluate which mutations are deleterious.

Using simulations, the authors show that GERP scores do not predict the strength of purifying selection well, turnover of functional elements and lineage-specific constraint complicates the interpretation of GERP scores, and most intermediate GERP scores may be sites affected by turnover.

Interestingly, simply increasing the number of species does not uniformly increase power if the simulation is done with turnover.

Based on mixture models, the authors estimate that 4.5% of the noncoding genome is under selection in human.

I have only a few comments:

1)

In the original paper, the GERP score is simply a measure of how many substitutions have been purged by purifying selection at a particular site. It is a measure of overall constraint, but not a direct measure of how deleterious a particular mutation is. Therefore, the authors should please define how the GERP score of a derived mutation is actually calculated (they refer to this several times in the introduction, e.g. page 6).

2)

This statement "It may be possible to reconcile these estimates by noting that they measure different processes—functional assays assess whether the nucleotide has a biological function, but this function may not necessarily be related to fitness (Graur et al. 2013; Doolittle 2013)"

Graur argues that biological function should be applied in the sense of which function was evolutionarily selected for. Biochemical activity as measured by Encode's 'functional assays' does often not imply a selected function.

3)

Page 20: Why not using the AIC criterion to evaluate whether added model complexity is justified?

4)

It would be interesting to explore the effect of adding more closely-related species. E.g. the authors could use the same data generated for Figure 4 but then restrict the analysis on the subtree of primates, where the amount of turnover is expected to be less than in the entire tree.

5)

I believe "or under selection" should be removed from this sentence

"For example, approximately 61.6% of GERP scores >5.5 in our simulations are from sites that are not functional or under selection in humans"

Reviewer #2: Huber et al. examined how one measure for assessing sequences under selection, Genomic Evolutionary Rate Profiling (GERP), is related to the population genetic strength of selection (NeS). They found: (1) they are related, (2) GERP distribution is impacted by changes in selection coefficient, or function over time, (3) more turnover in sequence elements’ functions correlates with smaller optimal tree size, (4) 4.5% of non-coding human genome is under purifying selection experiencing changes in selection coefficient over the course of mammalian evolution. One major question I have is with regard to the validity of simulation results in reflecting the mammalian evolutionary trajectory. I also have other questions that I hope the authors may find useful. This is a very long paper and I may have missed some of the points – but the authors may consider this point as a reader’ perspective – possibility of tightening the manuscript up some.

- For reader not familiar with the approach, the abstract’s mention of trees would be cryptic – need to provide info on the relevance. Also, spell out GERP in abstract.

- p.5, the authors commented on the use of conservation as a way to detect selection to be “[a] concept [that] given rise to the field of comparative genomics”. This is rather inaccurate, as comparative genomics is broader than looking at conservation and certainly the early evolution of the field was contributed heavily by non-evolutionary biologists.

- p.5, “A number of statistical approaches have been developed to find these sites … showing conservation” – provide citations.

- p.6, “GERP scores may not provide quantitative evidence of the strength of selection because any deleterious mutations that have a scaled selection coefficient of Nes < -2 will not accumulate as substitutions (Figure 1)” – this is intro section - perhaps talked about this as something that has not been evaluated and put the info in Figure 1 into results? Also, Figure 1A seems show a negative correlation. Will be helpful for the author to provide the GERP scores as distributions (e.g. violin plots) as it is hard to gauge what the central tendency of GERP values is.

- p.7, “Comparative genomic approaches assume that selective pressures have remained relative stable…” – it will be helpful to point out specific examples here, particularly using GERP as an example.

- p.8, “mutations at functional sites may not have an evolutionary impact and thus could appear to be neutral in comparative genomic approaches” – The authors need to be clear here. Given the interests of the manuscript is about detecting selection, it is not clear to me how these “functional” sites that have no evolutionary impact is relevant. Also, in this paragraph, the authors used the word “function” inconsistently. E.g. in the sentence following “sequences may have a biological function in some species and not others…” – here it seems that the “biological function” here is under selection, contradict with the statement “… whether the nucleotide has a biological function, but this function may not necessarily be related to fitness”. This is rather confusing.

- Figure 1 – I assume the approach for generate the result is based on the simulation of deleterious substitution along a phylogenetic tree discussed in p.30. The authors used HKY85, transition/transversion ratio of 2, and assume equilibrium frequencies to be equal for all four nucleotides. For the first two, would be helpful to know why they were picked and whether some parameter sweep was done to help with picking them out. For the equal equilibrium nucleotide frequency, given the human genome has a mean GC content of 41%, it seems unrealistic – can the authors explain why it was used? How do all these impact the discussion relevant to figure 1? How realistic is the resulted deleterious substitution simulation?

- p.10, “positive GERP scores are not very predictive of the strength of purifying selection on a particular variant” – this is based on Figure 1A, but the interpretation is problematic. It is true that a GERP score > 4 can be generated by nearly neutral mutations – the question is how often this happens. It is hard to tell from Figure 1A. It would have been helpful to know the percentile values. Besides, this is assuming that the deleterious simulation is realistic – which the authors should provide arguments that it is.

- Would it make more sense to infer GERP scores using simulated sequence data so the results are more directly comparable with the population genetic parameters used to generated the simulated sequences?

- Figure 2: instead of average, can the authors show median values (or violin/box plots) so one can get a better sense of the distribution? Can the authors use color scheme other than red-green for accessibility reason?

- NeS and NS are used interchangeably in figures, which can be confusing.

- p.12, “Many of the deleterious mutations show GERP score ….” – provide exact number, proportion to support claims.

- p.14, “Assuming functional turnover as outlined … results in a very different pattern” – looking at Figure 3, it is not clear if it would support that the pattern is “very different”. Although with turnover there is a lot more scatter, it seems that they can be attributed to outliers (as suggested by Figure 3B and the average line in Figure 3A). The authors discussed the most extreme situation (GERP>5.5) to say that ~62% are not “under selection” – but the criterion for calling selection (e.g. an NeS threshold) was not provided or justified.

- p.15 and on, on optimal tree size: One naïve notion in my mind is that, for functional turnover to have less impact on GERP, adding more species, and thus more resolution from a phylogenetic sense, would allow the approach to detect branches with turnover more readily. But the results in Figure 4 demonstrated otherwise. Wonder if the authors can explain this a bit.

- p.18, “We next fit a mixture distribution to the empirical GERP score distributions…” – looking into the methods, it was not clear how the mixture model is constructed – there is no mention of algorithm, parameters, variables. It is challenging to access what the significance is here.

- p.19, “We next assessed how well a model fits the data by examining the overlap of the distribution of standardized GERP scores observed in the data with the distribution of standardized GERP scores under a certain model” – the “certain model” here threw me off as I am not sure what it is. Looking into methods, while there are details how model overlaps were assessed, I am not sure where the “certain model” is described.

- p.22, estimate of the proportion of human genome under purifying selection – consider the assumptions going into the model (still not sure how it is constructed as pointed out above), how do they impact this estimate? Why is this estimate necessarily more accurate compared to earlier ones?

**Have all data underlying the figures and results presented in the manuscript been provided?**

Reviewer #1: Yes

Reviewer #2: Yes

PLOS authors have the option to publish the peer review history of their article (what does this mean?). If published, this will include your full peer review and any attached files.

Reviewer #1: No

Reviewer #2: No

---

## [Editor Report · Decision Letter 1]

5 May 2020

Dear Dr Lohmueller,

We are pleased to inform you that your manuscript entitled "Population genetic models of GERP scores suggest pervasive turnover of constrained sites across mammalian evolution" has been editorially accepted for publication in PLOS Genetics. Congratulations!

Yours sincerely,

Takashi Gojobori

Associate Editor

PLOS Genetics

Bret Payseur

Section Editor: Evolution

PLOS Genetics

Comments from the reviewers (if applicable):

**Data Deposition**

http://datadryad.org/submit?journalID=pgenetics&manu=PGENETICS-D-20-00130R1

**Press Queries**

---

## [Editor Report · Acceptance letter]

22 May 2020

PGENETICS-D-20-00130R1 

Population genetic models of GERP scores suggest pervasive turnover of constrained sites across mammalian evolution 

Dear Dr Lohmueller, 

We are pleased to inform you that your manuscript entitled "Population genetic models of GERP scores suggest pervasive turnover of constrained sites across mammalian evolution" has been formally accepted for publication in PLOS Genetics! Your manuscript is now with our production department and you will be notified of the publication date in due course.

With kind regards,

Matt Lyles

PLOS Genetics

On behalf of:
